# A duo-theme cloud model DEMATEL approach for exploring the cause factors of green supply chain management

**Jih-Kuang Chen**, **Tseng-Chan Tseng** *

Economics and Management College, Zhaoqing University, Zhaoqing, China

* 2018013044@zqu.edu.cn

**Data Availability Statement:** All relevant data are within the paper and its Supporting information files.

**Funding:** The authors received no specific funding for this work.

## Abstract

### Purpose

Decision-Making Trial and Evaluation Laboratory (DEMATEL) methods identify cause factors in green supply chain management (GSCM). This study argues that the target method treats affecting factors as unique themes; however, various factors may be mutually antagonistic (i.e., mutually positive or negative) or encompass other meaningful information (e.g., gain/risk, intensify/depress). The factor affecting GSCM implicitly encompasses the economy and ecology (greenness), which may conflict. This new approach can be integrated into the analysis, dividing affecting factors into "cause" and "effect" groups. The organization should focus on affecting factors in the cause group. The findings provide strategic guidance for organizations to practice GSCM.

### Design/Methodology/Approach

A duo-theme cloud model DEMATEL approach was proposed to divide these affecting factors of GSCM into "economy" and "greenness." The cloud model was applied to overcome the ambiguity and randomness in the concept of uncertainty and allow the integration of mutual qualitative and quantitative mapping.

### Findings

Six factors in the economic aspect and four in the greenness aspect should be classified as the cause group.

### Practical implications

Organizations should prioritize these ten factors in their GSCM practices. Doing so makes the GSCM problem relatively straightforward and allows for efficacious decision-making.

### Originality/Value

This study proposes a duo-theme cloud model DEMATEL approach to identify cause factors in GSCM.

**Competing interests:** No competing interests exist.

# 1. Introduction

Supply uncertainty causes significant economic losses, encompassing all processes transforming raw materials into final products (Lan et al., 2021) [1]. As the environment deteriorates and resources become increasingly scarce, the conflict between development and environmental protection is growing increasingly prominent. The essence of supply chain management has expanded into green supply chain management (GSCM), whose practices ideally would minimize environmental impacts and improve resource efficiency through all stages of the supply chain, from product procurement to final disposal of goods after use. GSCM helps organizations create "win-win situations" and balance economic and environmental benefits (Zhu and Sarkis, 2004) [2].

There have been many studies on this topic, including adopting and implementing several mathematical methods (Govindan et al., 2015) [3]. For example, researchers explored GSCM regarding the causality of influential factors. Most studies are based on the Decision-Making Trial and Evaluation Laboratory (DEMATEL) method, developed by the Battelle Memorial Institute in Geneva (Gabus and Fontela, 1973) [4]. The DEMATEL method reveals the relationships among influential factors based on relatively small amounts of data. DEMATEL creates a causal diagram of interdependent factors to visualize the relationships among these factors. However, some scholars argue that expert evaluations of the qualitative criteria of an object are always expressed linguistically in complex systems. However, such linguistic evaluations are vague and challenging to translate into crisp values. DEMATEL mixes all included factors as unique themes. Various factors may be mutually antagonistic (i.e., mutually positive or negative) or encompass other meaningful information (e.g., gain/risk, intensify/depress). The factors affecting GSCM implicitly encompass economy (e.g., product, production, income) and ecology (e.g., emissions reduction, green design, greenness), which may conflict.

This study proposed a duo-theme cloud model DEMATEL approach to divide factors affecting GSCM into "economy" and "greenness," and a cloud model was applied to overcome the ambiguity and randomness in the concept of uncertainty and allow integration of qualitative and quantitative mutual mapping. This approach can be integrated into the analysis, dividing affecting factors into "cause" or "effect" groups. Organizations should focus on the influencing factors in the cause group. The findings may provide strategic guidance for organizations to practice GSCM.

# 2. Literature review

Many studies used empirical research to explore the factors influencing GSCM, including green supplier levels, manufacturing efficiency, company activities, and environmental behavior of Malaysian manufacturing firms (Mohamed et al., 2020) [5]; another study explored the effect of green capabilities on GSCM adoption in Ghana (Nkrumah et al., 2021) [6]; yet another explored the causal relationships between the partnership governance mechanism and the success of GSCM practice (Lee and Choi, 2021) [7]. One study determined how a firm's relational capital of green and quality management in supply chains impacted its operational and environmental performance (Wu et al., 2020) [8]. Investigators identified what GSCM practices would impact business profitability for first-tier suppliers in the South Korean electronics industry (Park et al., 2022) [9]. Another study examined the environmental, social, and economic performance of green supply chain integration's influence on Chinese manufacturers' sustainable performance (Han and Huo, 2020) [10]. Investigators explored the relationship between green supply chain integration and firms' green innovation performance and its intrinsic mechanism of Chinese manufacturers (Zhang et al., 2022) [11]. Another study examined the impact of various lean manufacturing practices on sustainability performance and the

mediating role of GSCM for Pakistani manufacturing firms (Awan et al., 2022) [12]. Yet another group studied the relationship between corporate social responsibility, GSCM, and operational performance and the moderating effects of relational capital in China (Xu et al., 2022) [13]. Finally, a study revealed that public/supplier/competitor pressures drove GSCM practices in the Indian pharmaceutical supply chain (Sabat et al., 2022) [14].

Other studies used mathematical methods to explore GSCM. One study analyzed major factors and barriers in GSCM practice using an interpretive structural modeling approach (Singh et al., 2016) [15]; Fuzzy Preference Programming with Fuzzy VlseKriterijumska Optimizacija I Kompromisno Resenje was used to assess suppliers' performance with carbon management standard (Fallahpour et al., 2020) [16]. A study used a hybrid Entropy- technique for order of preference by similarity to an ideal solution (TOPSIS)-F approach to select the supplier with the best environmental performance for the Brazilian furniture industry (dos Santos et al., 2019) [17]. Another used an integrated fuzzy Best-Worst Method, Complex Proportional Assessment of Alternatives) and Weighted Aggregated Sum-Product Assessment evaluation for Iran's renewable energy supply chain (Masoomi et al., 2022) [18]. Another used fuzzy analytic hierarchy processes to calculate the weights of the supplier selection criteria for small and medium-sized enterprises (Buyukselcuk et al., 2022) [19]. Finally, a study integrated TOPSIS with a Cloud model to improve green supplier selection (Ramakrishnan and Chakraborty, 2020) [20].

DEMATEL and related methods have mushroomed in recent years, including Irajpour et al. (2012) [21], who assessed managerial and logistical factors to evaluate a green supplier. Verma and Gangele (2012) [22] investigated waste reduction and recycling processes for a pharmaceutical manufacturer in India. Wu & Chang (2015) [23] identified the critical dimensions and factors of GSCM for electrical and electronic industries in Taiwan. Rasi (2016) [24] developed a conceptual model for evaluating green suppliers based on DEMA-TEL. Fallahian-najafabadi et al. (2013) [25] evaluated the influence of factors among 22 criteria across five managerial factors. Mavi et al. (2013) [26] identified various logistical factors to evaluate a green supplier. Hsu et al. (2013) [27] utilized the DEMATEL method to recognize the influential carbon management criteria in a green supply chain. Lin et al. (2018) [28] developed the approximate fuzzy DEMATEL to analyze uncertain influential factors under the weakest t-norm arithmetic operations. Bai and Satir (2020) [29] applied Grey-DEMATEL and Grey-interpretive structural modeling to identify their relationships under uncertainty in the green supplier development practice. Liu et al. (2021) [30] used the Grey-DEMATEL method to examine the cause-effect relationships to reveal the drivers for second-tier supplier management. Mubarik et al. (2021) [31] applied the Grey-DEMATEL-ANP approach to identify the technology and environmental management system as the critical sub-criteria dimensions. Pourjavad and Shahin (2020) [32] integrated fuzzy DEMATEL, fuzzy AHP, and TOPSIS methods to investigate and prioritize green supplier development programs.

There have been many other valuable contributions to the literature. However, previous research on DEMATEL generally involved mutual influence assessment of the factors on unique themes; nevertheless, they must overcome the ambiguity and randomness in the concept of uncertainty. Therefore, a duo-theme cloud model DEMATEL approach is necessary. The proposed method is explained in detail in the next section.

## 3. Methodology

The duo-theme cloud model approach evaluated influential factors regarding economy and greenness. The procedure was as follows:

## 3.1 Standard cloud

The cloud model assumes that U is the quantized numeric field, and Ć is U's qualitative representation. In contrast, μ: U→ [0, 1], x → μ(x), ∀ x ∈ U, the degree of certainty for qualitative representation Ć is represented by quantitative numerical. The distribution of x over the quantized field U is called Cloud and expressed as C(x), where x is a set of quantitative representations. The Cloud model can transform between quantitative assessment and qualitative representations. It satisfies:

$$\mu(x) = \exp\left(-\frac{(x_i - Ex)^2}{2En_i^2}\right),$$

where:

$$x \rightarrow N(Ex, En^2), En \rightarrow N(Ex, He^2)$$

The cloud model has three digital features that form the key parameters. Expectation (Ex) is the expectation of the center of cloud droplets, which reflects the average. Entropy (En) represents the effective domain of U and maps the ambiguity. Hyperentropy (He) represents the degree of dispersion of assessment, which maps the thickness of the cloud droplets. The three key parameters are shown in Fig 1.

We can build standard clouds to reveal the extent of qualitative representation interaction. The degree of interaction can be divided into five levels: None, Lower, Middle, Higher, and Full (Table 1).

The characteristics of the standard cloud can be assessed by the following formula (1) (Wang and Zhu, 2012) [33], where k is a different adopted value for different studies, and 0.5 was adopted here, refer to Li et al. (2017) [34]:

$$\begin{cases} Ex_i = \frac{(d_i^{min} + d_i^{max})}{2} \\ En = \frac{(d_{max} - d_{min})}{6} \\ He = k \end{cases} \tag{1}$$

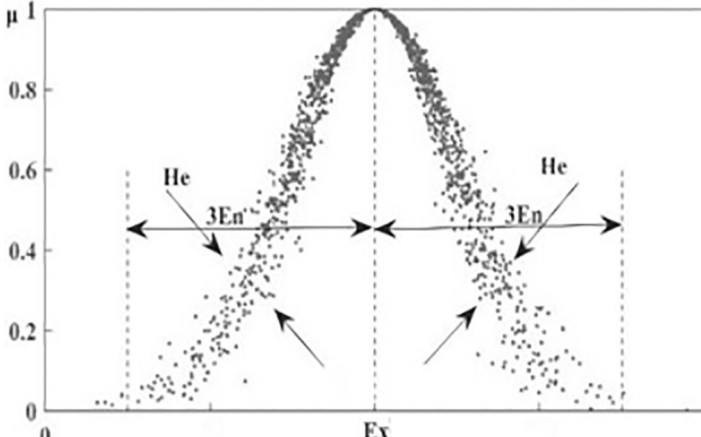

**Fig 1. The critical parameters of the cloud model.**

**Table 1. Degree of interaction and corresponding key parameters.**

| Degree of interaction | Linguistic terms | Value interval | Ex | En | He |
|---|---|---|---|---|---|
| None | 0 | [0, 0.8] | 0.4 | 0.133 | 0.5 |
| Low | 1 | [0.8, 1.6] | 1.2 | 0.133 | 0.5 |
| Middle | 2 | [1.6, 2.4] | 2 | 0.133 | 0.5 |
| Higher | 3 | [2.4, 3.2] | 2.8 | 0.133 | 0.5 |
| Full | 4 | [3.2, 4] | 3.7 | 0.133 | 0.5 |

### 3.2 Generating the digital characteristics

We can use a backward cloud generator to generate the digital features for a specific cloud droplet, as shown in Fig 2.

Applying a backward cloud generator, the three key parameters (Ex, En, He) of digital features reflect the mapping of cloud droplets from qualitative to quantitative. The following formula was used:

$$\begin{cases} \mathrm{Ex} = \sum_{i=1}^{n} x_i \Big/ n \\ \mathrm{En} = \sqrt{\frac{\pi}{2}} E(|x - Ex|) = \frac{1}{n} \sqrt{\frac{\pi}{2}} \sum_{i=1}^{n} |x_i - Ex| \\ \mathrm{He} = \sqrt{D(X) - En^2} = \sqrt{\frac{1}{n-1} \sum_{i=1}^{n} (x_i - Ex)^2 - En} = \sqrt{S^2 - En^2} \end{cases} \qquad (2)$$

### 3.3 Similarity comparison

Similarity Comparison is commonly used, and the Cloud Model-Based Similarity Comparison Method (LICM) (Zhang et al., 2007) [35] is a comparison between the three-dimensional vector (Cm) and a standard cloud. The LICM method uses the angle cosine of these vectors to define the similarity measure, where C1 and C2 are two 3D vectors, v1 and v2. The similarity measure of these two vectors is measured as shown in formula (3):

$$\mathrm{sim}(C_1, C_2) = \cos v_1 \cdot v_2 = \frac{v_1 \cdot v_2}{|v_1||v_2|} \qquad (3)$$

Finally, the backward cloud generator generated the similarity between the clouds, and the standard cloud was compared. The highest similarity corresponds to the closest assessment value, converted into corresponding linguistic terms to form the direct-relation matrix of DEMATEL. Finally, the total-relation matrix can be obtained by applying DEMATEL's operation rules.

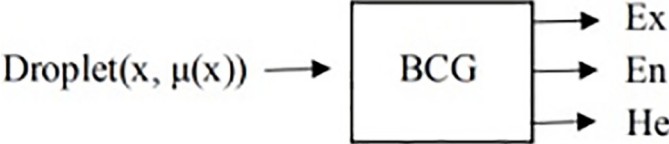

**Fig 2. Backward cloud generator.**

### 3.4 DEMATEL method

The direct-relation matrix $Z$ is composed of $z_{ij}^k$. DEMATEL is operated in the following stepwise process.

1. Normalization: The maximum value of the sum of the rows is taken as the normalized basis ($\lambda$) to calculate the normalized direct-relation matrix.

$$\lambda = \frac{1}{\max_{1 \leq i \leq n}\left(\sum_{j=1}^{n} z_{ij}\right)} \tag{4}$$

The direct-relation matrix $Z$ multiply by $\lambda$, then to obtain the normalized direct-relation matrix $N$:

$$N = \lambda \times Z \tag{5}$$

2. The total-relation matrix can be calculated based on the following formula, where I is the identity matrix:

$$T = \lim_{n \to \infty}(N + N^2 + \cdots N^K) = N(I - N)^{-1} \tag{6}$$

3. *Di* and *Rj* are calculated next. The *total-relation matrix calculates the Di and Rj values*, which include direct and indirect influences. $D_i$ is the sum of row $i$ and represents the sum for the cases where factor $i$ influences other factors; $R_j$ is the sum of column $j$ and represents the sum of the cases where factor $j$ is influenced by other factors.

$$\begin{cases} D_i = \sum_{i=1}^{n} t_{ij}(i = 1, 2, \ldots, n) \\ R_j = \sum_{j=1}^{n} t_{ij}(i = 1, 2, \ldots, n) \end{cases} \tag{7}$$

4. Next, to calculate the prominence (D+R) and the relation (D-R), D+R is defined as prominence representing the total degree of an element's influence and its ability to be influenced, i.e., the prominence of this element in the overall problem. D-R represents the extent to which this element is a cause or effect in all problems. If this value is positive, this element is a cause; if it is negative, it is an effect.

Many researchers have applied DEMATEL in many fields, but its algorithm has also been revised by researchers, such as WINGS (Jerzy, 2013) [36], multilayer hierarchical DEMATEL (Chen, 2022[①]) [37], and duo-theme DEMATEL (Lee and Wu, 2014) [38].

### 3.5 Duo-theme DEMATEL method

After completing the DEMATEL analysis on the economy and greenness aspects, the prominence value of one aspect's factor (e.g., "greenness") is then changed from positive to negative, and all factors are built into a comprehensive cause diagram. This process can be summarized

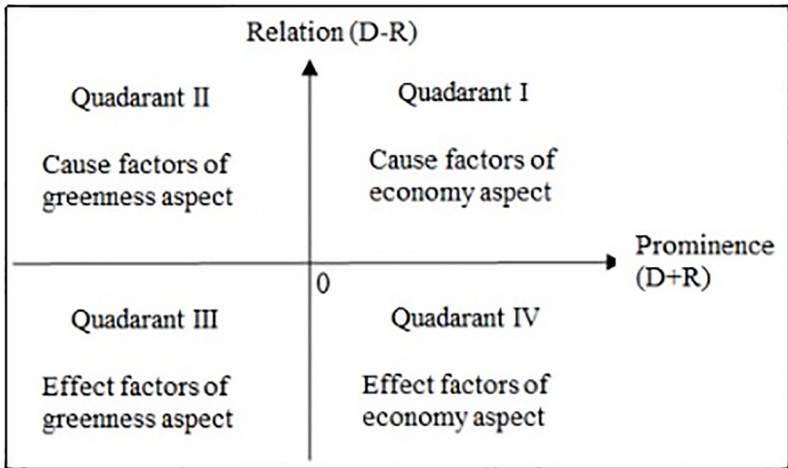

**Fig 3. Comprehensive causal diagram.**

as follows.

$$\begin{cases} \{X_i, Y_i\}_{economy} = \{(D_i + R_i), (D_i - R_i)\}, \text{ where i is i-th factor of the economy aspect} \\ \{X_j, Y_j\}_{greenness} = \{-(D_j + R_j), (D_j - R_j)\}, \text{ where j is j-th factor of greenness aspect} \end{cases} \quad (8)$$

As shown in Fig 3, a comprehensive causal diagram with economy-greenness aspects reveals that economy factors are in quadrants I and IV, and the greenness factors are in quadrants II and III. As described above, the D-R differentiates the cause from the effect groups. If the D-R is positive, the factor belongs to the cause group; if the D-R is negative, the factor belongs to the effect group. Researchers should focus on the factors in the cause group at quadrants I and II to optimize decision-making effects.

The cloud model dual-aspect DEMATEL method reflects the influential factors in GSCM, including the economy (ES) and greenness (GS) factors. The cause factors are crucial parts of the comprehensive diagram and should be prioritized to enhance GSCM practices. A summary of this procedure is illustrated in Fig 4.

## 4. Analysis

Based on the literature review, we identified 22 factors that influence GSCM practice; there are 12 economy factors and ten greenness factors. The factors that affect GSCM's business performance are divided into Economy factors. The factors that affect the environmental performance of SCM are classified as the Greenness factor. The affecting factors and symbol codes are displayed in Table 2.

Forty-three experts from the Guangdong-Hong Kong-Macao Greater Bay Area in China were invited, all of whom were industry experts in the management and implementation of GSCM for at least eight years or academic experts teaching GSCM-related courses and studies for at least eight years. Experts were invited to assess the interaction of influence factors of GSCM in enterprises in China's Guangdong-Hong Kong-Macao Greater Bay Area. Thirty-seven valid responses were obtained, of which 24 were from industry and 13 from academia. The valid responses were calculated by the cloud model to set up a direct-relation matrix.

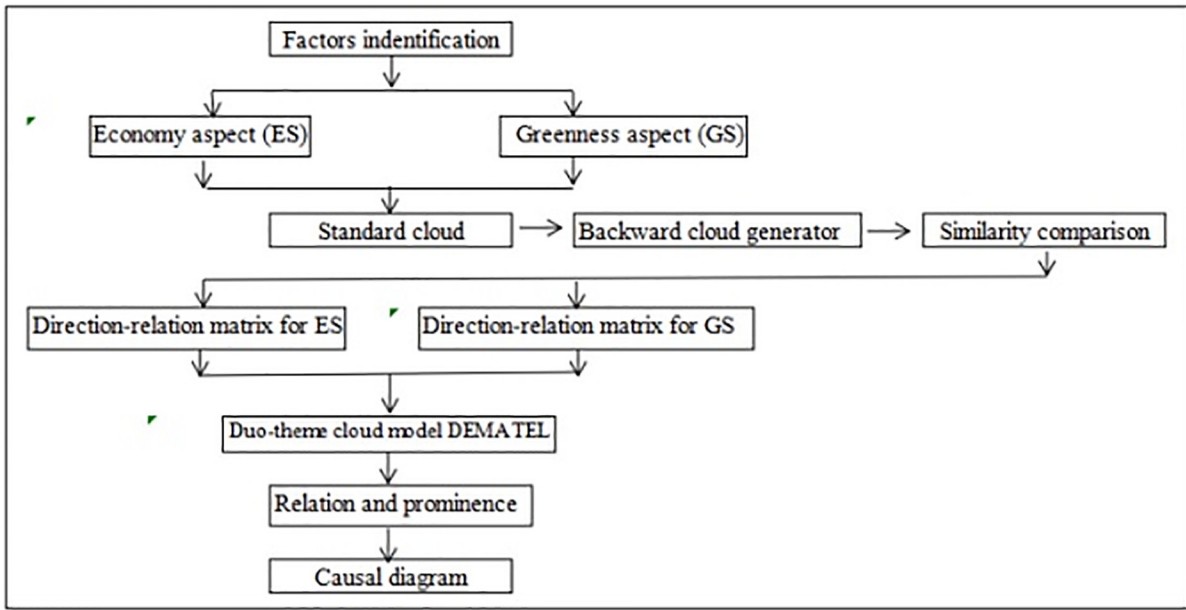

**Fig 4. Duo-theme cloud model DEMATEL analysis architecture.**

Respondents were asked to evaluate the influence of economic aspects and greenness aspects; scores of 0, 1, 2, 3, and 4 represent "None," "Low," "Middle," "Higher," and "Full," respectively. First, the standard clouds with 37 valid responses were calculated according to the instructions in Table 1 and Formula (1). Next, the backward cloud generator was used to generate the key parameters (Ex, En, and He) from the collected valid responses, according to Formula (2). The results are shown in Appendix I in S1 Appendix. Finally, the backward cloud generator generated the similarity between the clouds, and the standard cloud was compared. The highest similarity corresponds to the closest assessment value and transforms it into corresponding linguistic terms. The results are shown in Appendix II in S1 Appendix. The results of the two direct-relation matrixes, economy direct relation (Xe) and greenness direct relation (Xg), are shown in Appendix III in S1 Appendix.

**Table 2. The affecting factors of GSCM and symbols code.**

| Economy | Code | Greenness | Code |
|---|---|---|---|
| Market share | S1 | Capacity to adopt a green process | G1 |
| Product yield rate | S2 | Capacity to optimize the distribution of production resources | G2 |
| Punctual delivery rate | S3 | Selection of green materials | G3 |
| Clients retention rate | S4 | Green level of transport | G4 |
| Cash turnover | S5 | Input of energy conservation and emission reduction | G5 |
| Rate of return on total assets | S6 | Rate of environmental cost input | G6 |
| Information-sharing degree | S7 | Capacity of disposing of waste | G7 |
| R&D cycle of new product | S8 | Facility utilization | G8 |
| Accuracy of market forecast | S9 | Energy conservation rate | G9 |
| Rate of stock turnover | S10 | Resources reusing rate | GI0 |
| Response speed of supply chain | S11 | | |
| Production flexibility of supply chain | S12 | | |

**Table 3. Relation and prominence.**

| Economy aspect | Prominence (De+Re) | Relation (De-Re) | Greenness aspect | Prominence (Dg+Rg) | Relation (Dg-Rg) |
|---|---|---|---|---|---|
| S1 | 6.204 | 0.508 | G1 | -3.792 | -0.382 |
| S2 | 3.248 | 2.165 | G2 | -4.915 | -0.468 |
| S3 | 4.078 | -0.211 | G3 | -3.918 | 0.445 |
| S4 | 6.236 | -0.698 | G4 | -1.608 | 0.191 |
| S5 | 5.866 | -0.082 | G5 | -4.428 | 0.577 |
| S6 | 5.438 | -0.818 | G6 | -4.463 | 1.124 |
| S7 | 4.695 | 2.592 | G7 | -2.893 | -0.247 |
| S8 | 5.426 | 0.041 | G8 | -3.463 | -0.156 |
| S9 | 4.722 | 0.054 | G9 | -4.054 | -0.389 |
| S10 | 4.527 | -2.549 | G10 | -3.815 | -0.697 |
| S11 | 7.193 | -1.207 | | | |
| S12 | 5.582 | 0.204 | | | |

Then, the λ value was calculated as 1/19 for economy factors, and 1/21 for greenness factors per Eq (5). The direct-relation matrix Xe was then multiplied with the λ value to obtain the normalized direct-relation matrix Ne for economy factors and the normalized direct-relation matrix Ng for greenness factors, The results are shown in Appendix IV in S1 Appendix.

The normalized direct-relation matrix was used to calculate the total-relation matrix Te for economy factors and Tg for greenness factors using Eq (7). The results are shown in Appendix V in S1 Appendix.

After calculating the $D_i$ and $R_j$ values for each factor using Eq (8), the prominence (D+R) and relation (D-R) for each factor were calculated, as shown in Table 3.

Next, the coordinate points of economy and greenness factors were calculated using Eq (8) to obtain the duo-theme cloud model DEMATEL comprehensive cause diagram (Fig 5). The factors were divided into cause and effect groups, as can be observed in the diagram.

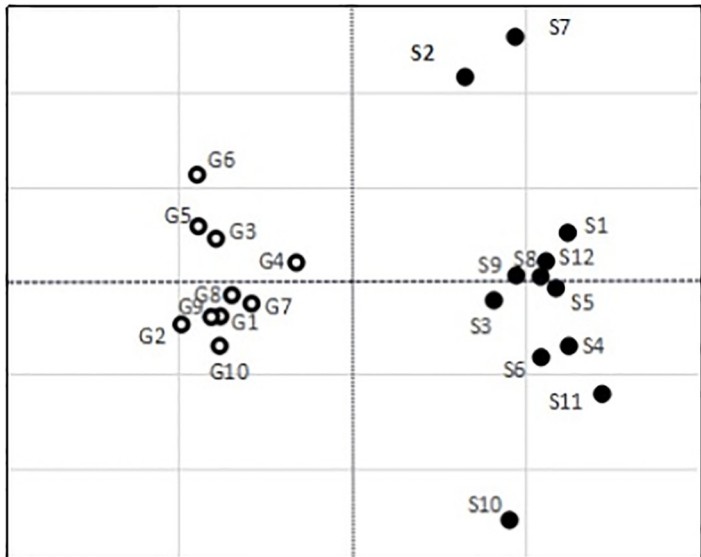

**Fig 5. Comprehensive causal diagram of duo-theme cloud model DEMATEL.**

## 5. Findings

### 5.1 Cause and effect factors of economy aspect

Economic cause factors have relation values greater than 0. The influence severities of these factors on other factors were determined based on the diagram as S7, S2, S1, S12, S8, and S9 in descending order of impact. These active factors have an economic influence on GSCM and should be prioritized by decision-makers. S7 has the most significant impact; this finding suggests that the degree of information-sharing is essential to each link in the supply chain. Organizations should optimize the connections among supply and demand agents to improve the speed of responses. Accurate information-sharing can facilitate effective management, reduce costs, and increase resource utilization. S2 is the second-most critical factor, suggesting that product yield rate is also critical. Product quality is critically critical to the organization as it fundamentally affects profits. The third-most critical factor is S1; a loss in market share results if the product yield rate is not sufficiently high. S12, production flexibility, is another critical causal factor that describes the ability of suppliers to adjust the general output level to meet the demands of clients.

Economic effect factors are factors with relation values smaller than 0; other factors primarily influence these factors. We identified S10, S11, S6, S4, S9, S3, and S5 as the economic effect factors in descending order of their relation values. Among them, the stock turnover rate is primarily influenced by other factors, including cash turnover and the response speed of the supply chain. In other words, the stock turnover rate can be improved by adjusting these two factors primarily.

Our analysis of the least-most critical causal factors, S8 and S9, suggests that the traditional supply chain is efficiency-based and mainly centered on the mass production stage. However, without development and design cycles for new products and accurate market forecasts, the interactions of the supply chain may limit product development or negatively affect subsequent interactions with suppliers.

### 5.2 Cause and effect factors of greenness aspect

Green causal factors with relation values greater than 0 include G6, G5, G3, and G4. These factors actively influence the GSCM and should be highly prioritized. Both environmental impact and resource efficiency are considered in the whole supply chain of suppliers, producers, dealers, and users. Adverse environmental effects can be minimized, and resource efficiency maximized throughout obtaining materials, processing, packaging, storage, transport, usage, and discarding of the product. G6 showed the highest impact among green causal factors in this analysis. This suggests that the efficient utilization of environmental cost input is of great importance to GSCM practice. The relation value of G5 is the second-highest, indicating that enterprises should focus on energy conservation and emissions reduction. G3 ranks third, indicating that enterprises should start at the research and development stage to maximize greenness fully. The most efficient way is to select green materials for various components. Adopting green materials continually enhances recovery during the processing or reusing of materials for subsequent products. G4's relation value is smaller than G3, though transport is a critical link in GSCM. The transport from raw material factories to processing plants and from processing plants to the locations where products may be costly and have adverse environmental effects.

Green effect factors with relation values smaller than 0 include G10, G2, G9, G1, G7, and G8, in descending order of absolute value. Other factors influence these factors to affect GSCM practices. Among them, the resource reuse rate is intensely influenced by other factors,

including the rate of environmental cost input and selection of green materials; the resource reuse rate can be improved by adjusting these two factors.

Enterprises should optimize greenness and focus on developing an integrated transport system for their supply chains. This could start by increasing development in an energy-conserving manner, reducing and eliminating the use of older and less energy-efficient vehicles, and improving the overall level of energy conservation and environmental protection for transport vehicles, ports, and stations. The salient economic utilization of resources and effective planning of facilities for the supply chain can also reduce overall transportation needs and subsequently reduce transportation costs and resources.

## 6. Conclusions

This study proposed a duo-theme cloud model DEMATEL approach to identify affecting factors for successful GSCM practices. The proposed approach can divide these affecting factors of GSCM into "economy" and "greenness." The cloud model was applied to overcome the ambiguity and randomness in the concept of uncertainty and allow the integration of qualitative and quantitative mutual mapping. This approach can be further integrated into the analysis, dividing influential factors into "cause" or "effect" groups. This makes the GSCM problem relatively straightforward and allows for efficacious decision-making.

Several managerial implications can be derived based on the findings presented here. In practice, the factors in the cause group are more effective than those in the effect group; therefore, the factors in the cause group should be given priority. The causal relationships across all factors can be identified by drawing a comprehensive causal diagram according to $D_i$ and $R_j$ values calculated from the total-relation matrix. Organizations should prioritize ten factors in their GSCM practices: information-sharing degree, product yield rate, market share, supply chain production flexibility, new product R&D cycles, the accuracy of the market forecast, the rate of environmental cost input, energy conservation input, green material selections, and green level of transport.

However, there may be significant differences between different industries and regions, resulting in different influencing factors, so the conclusions obtained in this study may not be fully applicable. It is suggested that this method can be used to analyze another industry or region again, and the conclusions obtained will be more applicable to the analyzed industry or region. Future work should also determine the hierarchical structure of critical factors in GSCM using DEMATEL-ISM (Interpretive Structural Modeling) (Chen, 2022[②]) [39].

## Supporting information

**S1 Appendix.**
(DOCX)

**S1 File.**
(XLSX)

## Author Contributions

**Conceptualization:** Jih-Kuang Chen.

**Formal analysis:** Jih-Kuang Chen.

**Investigation:** Tseng-Chan Tseng.

**Methodology:** Jih-Kuang Chen.

**Supervision:** Tseng-Chan Tseng.

**Validation:** Tseng-Chan Tseng.

**Writing – original draft:** Jih-Kuang Chen.

**Writing – review & editing:** Tseng-Chan Tseng.

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
