## [Decision Letter · Decision Letter 0]

10 Sep 2023

PONE-D-23-24674A duo-theme cloud model DEMATEL approach for exploring the cause factors of green supply chain managementPLOS ONE

Dear Dr. Tseng,

Thank you for submitting your manuscript to PLOS ONE. After careful consideration, we feel that it has merit but does not fully meet PLOS ONE’s publication criteria as it currently stands. Therefore, we invite you to submit a revised version of the manuscript that addresses the points raised during the review process.

We look forward to receiving your revised manuscript.

Kind regards,

Baogui Xin, Ph.D.

Academic Editor

PLOS ONE

Journal Requirements:

   "NO"

4. Please ensure that you refer to Figure 3 in your text as, if accepted, production will need this reference to link the reader to the figure.

5. We note you have included a table to which you do not refer in the text of your manuscript. Please ensure that you refer to Table 3-7 in your text; if accepted, production will need this reference to link the reader to the Table.

Additional Editor Comments:

I recommend that it should be revised taking into account the changes requested by the reviewers. Since the requested changes include valuable and constructive reviews, I would like to give you a chance to revise your manuscript. The revised manuscript will undergo the next round of review by same reviewers.

Reviewers' comments:

Reviewer's Responses to Questions

**Comments to the Author**

1. Is the manuscript technically sound, and do the data support the conclusions?

Reviewer #1: Yes

Reviewer #2: Partly

2. Has the statistical analysis been performed appropriately and rigorously? 

Reviewer #1: Yes

Reviewer #2: Yes

3. Have the authors made all data underlying the findings in their manuscript fully available?

Reviewer #1: Yes

Reviewer #2: Yes

4. Is the manuscript presented in an intelligible fashion and written in standard English?

Reviewer #1: Yes

Reviewer #2: Yes

5. Review Comments to the Author

Reviewer #1: This manuscript showed a duo-theme cloud model DEMATEL approach for exploring the cause factors of green supply chain management. This new approach can be integrated into the analysis, dividing affecting factors into “cause” or “effect” groups. The cloud model was applied to overcome the ambiguity and randomness in the concept of uncertainty and allow the integration of mutual qualitative and quantitative mapping.

The following major points should be addressed:

(1) Centered figure and table names.

(2) Reference format improvement.

(3) The writing of English should be seriously improved.

(4) The DEMATEL method has been questioned and revised by scholars, please refer to and consider whether it is placed in the research limitation, such as:

Jerzy, M. (2013). Weighted Influence Non-linear Gauge System (WINGS) – An analysis method for the systems of interrelated components.

(5) Method : The dual-aspect FDEMATEL method? Should be duo-theme cloud model DEMATEL approach.

(6) The theoretical implication is clear, but the practical implication should be clarified more clearly.

(7)The supplementary discussion part explains the limitations of the method and looks forward to the future.

Reviewer #2: (1) Decision-Making Trial and Evaluation Laboratory (DEMATEL) methods are often used to

identify the cause factors of green supply chain management (GSCM). It is necessary to define the object of green supply chain management, and it is recommended to combine specific cases, Now too macro.

(2) It is necessary to indicate the criteria selected by the experts surveyed and Add credibility to the conclusions.

(3) How are indicators defined(economy and ecology).

(4) You can put the data in the appendix, and add the method description to the body.

6. PLOS authors have the option to publish the peer review history of their article (what does this mean?). If published, this will include your full peer review and any attached files.

Reviewer #1: No

Reviewer #2: No

---

## [Author Response · Author response to Decision Letter 0]

8 Oct 2023

Dear reviewer #1,

Thank you very much for your positive and constructive comments and suggestions, we have studied your comments carefully and have revised the manuscript. The amendments are highlighted in red in the revised manuscript. 

Following are the responds for your comments:

Response:

(1)All figure and table names have been centered.

(2)References have been proofread and appropriately formatted.

(3)A native English speaker proofread the entire manuscript.

(4)Added to reference.

(5)It is a typing error that has been corrected.

(6)The practical implication has been enhanced.

(7)Explanation has been enhanced to research limitations and future research direction.

Note: All revisions are indicated in red font. 

Dear reviewer #2,

Thank you very much for your positive and constructive comments and suggestions, we have studied your comments carefully and have revised the manuscript. The amendments are highlighted in blue in the revised manuscript. 

Following are the responds for your comments:

Response:

(1)The explanation has been enhanced to illustrate the research object and describe specific cases.

(2)Explanations have been enhanced regarding what criteria the experts select.

(3)An enhanced explanation has been provided to illustrate the definitions of both.

(4)The data of the calculation process has been moved to the appendix; the relevant method description has been enhanced.

Note: All revisions are indicated in blue font.

---

## [Decision Letter · Decision Letter 1]

7 Nov 2023

A duo-theme cloud model DEMATEL approach for exploring the cause factors of green supply chain management

PONE-D-23-24674R1

Dear Dr. Tseng,

We’re pleased to inform you that your manuscript has been judged scientifically suitable for publication and will be formally accepted for publication once it meets all outstanding technical requirements.

Kind regards,

Baogui Xin, Ph.D.

Academic Editor

PLOS ONE

Additional Editor Comments (optional):

Reviewers' comments:

Reviewer's Responses to Questions

**Comments to the Author**

1. If the authors have adequately addressed your comments raised in a previous round of review and you feel that this manuscript is now acceptable for publication, you may indicate that here to bypass the “Comments to the Author” section, enter your conflict of interest statement in the “Confidential to Editor” section, and submit your "Accept" recommendation.

Reviewer #1: All comments have been addressed

2. Is the manuscript technically sound, and do the data support the conclusions?

Reviewer #1: No

3. Has the statistical analysis been performed appropriately and rigorously? 

Reviewer #1: Yes

4. Have the authors made all data underlying the findings in their manuscript fully available?

Reviewer #1: Yes

5. Is the manuscript presented in an intelligible fashion and written in standard English?

Reviewer #1: Yes

6. Review Comments to the Author

Reviewer #1: These authors responded to the modification suggestions and there are not reviewer's Comments to the Author. It is recommended to accept.

7. PLOS authors have the option to publish the peer review history of their article (what does this mean?). If published, this will include your full peer review and any attached files.

Reviewer #1: No

---

## [Editor Report · Acceptance letter]

17 Nov 2023

PONE-D-23-24674R1 

A duo-theme cloud model DEMATEL approach for exploring the cause factors of green supply chain management 

Dear Dr. Tseng:

I'm pleased to inform you that your manuscript has been deemed suitable for publication in PLOS ONE. Congratulations! Your manuscript is now with our production department. 

Kind regards, 

on behalf of

Professor Baogui Xin 

Academic Editor

PLOS ONE